# Renal Endocytic Regulation of Vitamin D Metabolism during Maturation and Aging in Laying Hens

**DOI:** 10.3390/ani14030502

**Published:** 2024-02-02

**Authors:** Nami Kuwata, Hatsune Mukohda, Hiroto Uchida, Ryo Takamatsu, Muhammet Mustafa Binici, Takahisa Yamada, Toshie Sugiyama

**Affiliations:** 1Graduate School of Science and Technology, Niigata University, 2-8050 Ikarashi, Nishi-ku, Niigata 9502181, Japan; 2Department of Animal Science, Faculty of Agriculture, Niigata University, 2-8050 Ikarashi, Nishi-ku, Niigata 9502181, Japan; tyamada@agr.niigata-u.ac.jp

**Keywords:** vitamin D, megalin, cubilin, endocytosis, laying hen

## Abstract

**Simple Summary:**

Egg-laying hens have a specific and dramatic calcium metabolism to lay eggs with eggshells that are composed of calcium carbonate. Calcium metabolism is mainly regulated by vitamin D_3_. However, details of the regulatory mechanism of vitamin D_3_ metabolism have not been clarified. In this study, we found that the 25-hydroxyvitamin D_3_-binding protein complex is taken up by the endocytic receptors megalin and cubilin in the renal proximal tubules, providing a novel mechanism for the regulation of vitamin D_3_ metabolism during maturation and aging. These findings may contribute to the prevention of heat stress and age-related deterioration of eggshell quality in egg-laying hens.

**Abstract:**

Egg-laying hens undergo a specific and dramatic calcium metabolism to lay eggs with eggshells composed of calcium carbonate. Calcium metabolism is mainly regulated by vitamin D_3_. Although vitamin D_3_ metabolism is closely related to the deterioration of eggshell quality associated with aging and heat stress, the details of the mechanisms regulating vitamin D_3_ metabolism are not clear. In mammals, the vitamin D_3_ metabolite (25(OH)D_3_) produced in the liver binds to the vitamin binding protein (DBP), is subsequently taken up by renal proximal tubular cells via the endocytic receptors megalin (Meg) and cubilin (CUB), and is metabolized to 1,25-dihydroxyvitamin D_3_ (1,25(OH)_2_D_3_). Therefore, the present study aimed to examine the expression and localization of Meg and CUB in the kidneys of immature chicks and mature and aged laying hens to prevent eggshell quality deterioration. As a result, we showed that as circulating 1,25(OH)_2_D_3_ concentrations increased from 156.0 ± 13.5 pg/mL to 815.5 ± 61.4 pg/mL with maturation in immature chicks, relative expression levels (arbitrary units; AU) of *Meg* and *CUB* mRNA in the kidneys of mature hens significantly increased 1.92- and 2.75-fold, respectively, compared to those in immature chicks. On the other hand, the *Meg* mRNA expression levels of mature hens did not change with age, while *CUB* mRNA expression levels (1.03 ± 0.11 AU) were significantly decreased compared to mature hens (2.75 ± 0.24 AU). Immunohistochemical observations showed that Meg and CUB proteins were localized to the apical membrane of renal proximal tubular epithelial cells in immature chicks, mature hens, and aged hens, and that DBP protein was observed as granular endosomes in the cytoplasm of proximal tubular cells from the apical membrane to the cell nucleus. Especially in mature hens, the endosomes were larger and more numerous than those in immature chicks. In contrast, in aged hens, DBP-containing endosomes were smaller and limited to the apical cytoplasm. These results indicate that with maturation, the expression of Meg and CUB is promoted in the renal proximal tubules of laying hens, facilitating the uptake of the 25(OH)D_3_-DBP complex and its conversion to 1,25(OH)_2_D_3_, and regulating calcium metabolism in eggshell formation. On the other hand, it is suggested that the age-related decrease in CUB expression suppresses the uptake of the 25(OH)D_3_-DBP complex in the kidney, resulting in a deterioration of eggshell quality.

## 1. Introduction

Aging and heat stress in laying hens lead to the deterioration of eggshell quality, causing significant economic losses in the egg industry [1,2]. A large part of the eggshell (95%) is composed of calcium carbonate (CaCO_3_); thus, laying hens require large amounts of calcium for eggshell formation and have a specific calcium metabolism that differs from that of mammals [3,4]. The two main sources of calcium for eggshell formation are dietary calcium absorbed from the intestinal tract and calcium released from the medullary bone, which is a specific tissue in mature avian females. These eggshell-forming calcium metabolisms are considered to be regulated primarily by the active hormonal vitamin D_3_ metabolite (1,25-dihydroxy vitamin D_3_; 1,25(OH)_2_D_3_), one of the major calcium-regulating hormones, and it has been shown that circulating 1,25(OH)_2_D_3_ levels increase with the onset of egg laying and decrease with age [3,4,5]. It has also been shown that the 1,25(OH)_2_D_3_ levels are closely related to eggshell quality [6]. However, the mechanisms underlying this metabolic regulation remain unclear.

Vitamin D_3_ is synthesized in the skin from 7-dehydrocholesterol by ultraviolet irradiation in sunlight or is supplemented directly from dietary sources [7]. Vitamin D_3_ is transported to the liver and converted to 25-hydroxy vitamin D_3_ (25(OH)D_3_) by 25-hydroxylase (*Cyp27A1*), which then binds to vitamin D-binding protein (DBP), circulates in the blood, is taken up by the kidneys, and is converted to 1,25(OH)_2_D_3_ by 25(OH)D_3_-1α hydroxylase (*Cyp27C1* in birds, *Cyp27B1* in mammals) [8,9,10]. 1,25(OH)_2_D_3_ is biologically active and binds to vitamin D receptors to promote intestinal absorption and renal reabsorption of calcium, as well as the formation of medullary bone to maintain circulating calcium levels in the blood and promote eggshell formation in the oviduct [11,12]. Excessive 1,25(OH)_2_D_3_ production also results in loop feedback, which converts 25(OH)D_3_ to 24,25-dihydroxy vitamin D_3_ or 1,25(OH)_2_D_3_ to 1,24,25-trihydroxy vitamin D_3_ via 24-hydroxylase (*Cyp24A1*) in the kidney, thereby suppressing 1,25(OH)_2_D_3_ synthesis [7].

In mammals, the circulating 25(OH)D_3_-DBP complex has been shown to be taken up by endocytosis in renal proximal tubular cells via the endocytosis receptors megalin and cubilin [13,14]. Megalin (Meg) is a 600 kDa membrane receptor protein that functions as a ligand for a wide variety of substances, including low-density lipoprotein (LDL), DBP, and steroid hormones, and in taking them into the cell [15,16]. Similarly, cubilin (CUB), a member of the LDL receptor family, is a 460 kDa membrane protein, also known as the intrinsic factor vitamin B12 receptor in the small intestine [17]. Meg and CUB colocalize at the brush border of epithelial cells. Because CUB lacks transmembrane and cytoplasmic domains, it requires interactions with other membrane proteins during endocytosis [18]. Kidney studies in Meg-deficient mice and CUB-deficient dogs have shown that the 25(OH)D_3_-DBP complex promotes endocytosis more efficiently through interaction with Meg by CUB than by Meg alone [13].

In birds, the *Meg* gene has also been cloned, and its expression is prominent in proximal renal tubules. Its expression is higher in mature females and increases in an estrogen-dependent manner [19]. These results indicate that Meg plays an important role in LDL reabsorption in laying hens [20].

Therefore, in this study, we first determined the mRNA and protein expression of Meg and CUB in various tissues of mature laying hens to understand the mechanism of vitamin D_3_ metabolism via endocytosis in mature hens. Furthermore, to elucidate the regulatory mechanisms of vitamin D_3_ metabolism, we examined changes in the expression and localization of Meg and CUB in the kidney during maturation and aging, as well as the expression of enzymes related to vitamin D_3_ metabolism.

## 2. Materials and Methods

The animal experimentation protocol for the care of chicks and hens, as well as all experimental procedures in this study, were approved by the Committee for Animal Experiments at Niigata University (reference number SA00493, SA01395).

### 2.1. Animals and Experimental Design

For this study, 20 immature (10 days old), mature (330 days old), and aged (600 days old) White Leghorn hens were kept at the Niigata University Animal Experimentation Facility. All chicks and hens were given free access to water and standard commercial diets. The levels of calcium, phosphorus, vitamin D_3_ (2000 IU/kg diet for immature chicks and 3300 IU/kg diet for laying hens), and other nutrients in both diets satisfied the NRC standard requirements for laying hens. Mature and aged hens were fed the same diet, starting from their first egg laying. Seven chicks and mature and aged hens were randomly selected from them for this experiment. Mature hens, when an egg was in the shell gland of the oviduct (12 h after oviposition), were euthanized using carbon dioxide, and samples of the kidney, liver, duodenum, jejunum, ileum, magnum, isthmus, shell gland, and femur medullary bone were immediately excised [21]. Similarly, the kidneys and livers were immediately excised from chicks and aged hens. Each excised tissue was quick-frozen in liquid nitrogen and stored at −80 °C for quantitative PCR and western blotting analyses. Some excised tissues were fixed in 10% buffered formalin for 2 days at 4 °C and stored in 70% ethanol at 4 °C for immunohistochemical analysis of Meg, CUB, and DBP. Blood was collected and centrifuged at 800× *g* for 15 min to obtain serum.

### 2.2. Determination of Calcium and 1,25(OH)_2_D_3_ Concentrations in Serum

The serum concentrations of total calcium (mg/dL) and 1,25(OH)_2_D_3_ (pg/mL) were assayed using the Calcium E-Test Wako (Fujifilm Wako Pure Chemical Co., Osaka, Japan) and 1,25(OH)_2_ D RIA kit FR (Fujirebio Inc., Tokyo, Japan), respectively, according to the manufacturer’s instructions.

### 2.3. Total RNA Extraction, cDNA Synthesis, Quantitative-PCR Analyses

Total RNA was extracted using ISOGEN II (Nippon Gene, Tokyo, Japan), and cDNA was synthesized from total RNA using ReverTra Ace qPCR RT Master Mix (Toyobo Co., Ltd., Osaka, Japan) according to the manufacturer’s instructions. Quantitative PCR (qPCR) was conducted using the THUNDERBIRD^®^ Next SYBR^®^ qPCR Mix with gDNA Remover (Toyobo). The PCR cycling conditions were as follows: initial denaturation for 30 s at 95 °C; denaturation for 30 s at 95 °C; annealing and extension for 1 min at 60 °C; PCR was performed in 40 cycles. The relative expression level of each gene mRNA was calculated by the ΔΔCT method and normalized by β-actin. Values are expressed in arbitrary units (AU) relative to β-actin [9]. The primer sequences used are listed in Table 1.

### 2.4. Protein Extraction and Western Blotting Analyses

Quick-frozen tissues were homogenized in RIPA Lysis buffer (ATTO Co., Tokyo, Japan) at a concentration of 100 mg tissue/mL. The resulting total tissue lysate was used for protein determination using the BCA Protein Assay Kit (Takara Bio Inc., Shiga, Japan), and the remainder was dissolved in SDS-PAGE sample buffer and heated to 95 °C for 5 min. Protein samples (10 μg protein/lane) were electrophoresed on a 5% polyacrylamide gel (Bio-Rad Laboratories Inc., Hercules, CA, USA). Proteins from SDS-PAGE were semi-dry transferred onto PVDF membranes (Bio-Rad) for 50 min at 500 mA, constant at 15 V, using a Trans-Blot Turbo (Bio-Rad). EzFastBlot HMW (ATTO) was used as the transfer buffer for high molecular weight blotting. Immunoblots were probed with rabbit anti-Meg polyclonal antibody (Abcam plc, Cambridge, UK) (dilution 1:800) or sheep anti-CUB polyclonal antibody (R&D Systems, Inc., Minneapolis, MN, USA) (dilution 1:400) overnight at 4 °C as the primary antibodies. After washing with TBS-T, the secondary antibodies, HRP-conjugated goat anti-rabbit IgG polyclonal antibody (Abcam) (dilution 1:20,000) for Meg and HRP-conjugated rabbit anti-sheep IgG polyclonal antibody (Abcam) (dilution 1:4000) for CUB, were applied for an hour at room temperature. Immunoreactions were detected using ImmunoStar LD (Fujifilm Wako Pure Chemical Co.) and observed with a Lumino Image Analyzer LAS 3000 (Fuji Film Co., Ltd., Tokyo, Japan).

### 2.5. Immunohistochemistry

After fixation, the tissues were dehydrated using graded ethanol and xylene, embedded in paraffin wax, and sectioned at 5 μm. Immunohistochemical detection was performed using the avidin-biotin complex (ABC) method (VECTASTAIN ABC-AP Standard Kit; Vector Laboratories, Inc., Burlingame, CA, USA). The sections were treated for antigen retrieval in an immunosaver (Nissin EM Corporation, Tokyo, Japan) (dilution 1:200) for 45 min at 98 °C. Non-specific endogenous alkaline phosphatase activity was eliminated using BLOXALL^®^ Endogenous Blocking Solution (Vector Laboratories) for 10 min at room temperature. Background staining was blocked by incubating the sections with 1% normal serum of secondary antibodies in PBS for 20 min at room temperature. As the primary antibodies, rabbit anti-Meg polyclonal antibody (Abcam) (dilution 1:800) was applied for 30 min at room temperature, and sheep anti-CUB polyclonal antibody (R&D Systems) (dilution 1:400) was applied overnight at 4 °C. Alternatively, rabbit anti-DBP polyclonal antibody (Proteintech, Inc., Rosemont, IL, USA) (dilution 1:400) was applied for 1 h and 30 min at room temperature. The sections were washed in PBS, and then the secondary antibodies, goat anti-rabbit IgG polyclonal antibody biotinylated (Vector Laboratories) for Meg and DBP, or rabbit anti-sheep IgG polyclonal antibody biotinylated (Vector Laboratories) for CUB, were applied for 30 min at room temperature. Thereafter, the sections were incubated with alkaline phosphatase (AP)-conjugated ABC (Vector Laboratories) for 30 min. The AP reaction product was developed using ImmPACT Vector Red Substrate (Vector Laboratories). The sections were rinsed in distilled water, counterstained with Mayer’s hematoxylin solution for 45 s, and dehydrated in a graded series of ethanol and xylene before being placed on a coverslip for evaluation by light microscopy.

### 2.6. Statistical Analyses

All values are presented as the mean ± standard error (SE). Statistical analyses were performed using one-way ANOVA followed by Neuman-Keuls post-hoc analysis using GraphPad Prism 10 (GraphPad Software Inc., Boston, MA, USA). The significance level was set at *p* < 0.05.

## 3. Results

### 3.1. Serum Calcium and 1,25(OH)_2_D_3_ Concentrations

Serum total calcium concentrations were 11.3 ± 0.4 mg/dL in immature chicks, 30.4 ± 1.4 mg/dL in mature laying hens, and 33.5 ± 2.8 mg/dL in aged hens, and were significantly higher in mature and aged hens than in immature chicks (Figure 1A). Similarly, serum 1,25(OH)_2_D_3_ concentrations were 156.0 ± 13.5 pg/mL in immature chicks, 815.5 ± 61.4 pg/mL in mature hens, and 610.3 ± 96.5 pg/mL in aged hens, significantly higher than that in immature chicks (Figure 1B). Compared to mature hens, aged hens exhibited a significant decrease in serum 1,25(OH) _2_D_3_ concentration.

### 3.2. Relative Expression Levels of Meg and CUB mRNAs in Various Tissues of Mature Laying Hens by qPCR

The relative expression level (AU) of *Meg* mRNA was 1.0000 ± 0.1651 in the kidney, 0.0014 ± 0.0006 in the liver, 0.0056 ± 0.0008 in the magnum, 0.0016 ± 0.0003 in the isthmus, 0.0009 ± 0.0001 in the shell gland, and 0.0092 ± 0.0060 in the medullary bone (Figure 2A). The expression level of *Meg* mRNA in the kidney was extremely high compared with that in other tissues, similar to a previous report [19]. However, *Meg* mRNA expression levels in the intestinal tract (duodenum, ileum, and jejunum) were almost nonexistent. Similarly, the relative expression levels (AU) of *CUB* were 1.0000 ± 0.1399 in the kidney, 0.0021 ± 0.0003 in the duodenum, 0.0135 ± 0.0025 in the jejunum, and 0.0007 ± 0.0003 in the ileum, indicating that *CUB* mRNA was highly expressed in the kidneys compared with other tissues (Figure 2B). *CUB* mRNA expression was also observed in the duodenum, jejunum, and ileum, although to a lesser extent.

### 3.3. Expression of Meg and CUB Proteins in Various Tissues of Mature Laying Hens by Western Blotting

The extracted proteins were used to examine the expression of Meg and CUB proteins using western blotting. A rabbit anti-α-tubulin polyclonal antibody (Proteintech) was used as a loading control. A single band was observed in the kidneys at 600 kDa for Meg (Figure 3A) and 460 kDa for CUB (Figure 3B). No protein expression was observed in other tissues, indicating high protein expression in the kidneys as protein levels (Figure 3A,B).

### 3.4. Relative Expression Levels of Meg, CUB and Vitamin D Metabolism-Related Enzyme mRNAs in Immature Chicks and Laying Hens by qPCR

The relative expression levels (AU) of *Meg* mRNA in kidneys were 1.00 ± 0.13 in immature hens, 1.92 ± 0.26 in mature hens, and 2.63 ± 0.29 in aged hens, showing a significant increase with maturity and aging (Figure 4A), as described in the previous report [19]. The relative expression levels (AU) of *CUB* mRNA in kidneys were 1.00 ± 0.14 in immature chicks, 2.75 ± 0.24 in mature hens, and 1.03 ± 0.11 in aged hens, showing significantly higher expression levels in mature hens compared to immature chicks but significantly decreased in aged hens, with expression levels similar to those in immature chicks (Figure 4B).

The relative expression levels (AU) of *Cyp27C1* mRNA in the kidneys were 1.00 ± 0.20 in immature chicks, 2.89 ± 0.80 in mature hens, and 1.63 ± 0.27 in aged hens, with mature hens showing significantly higher expression levels than immature chicks (Figure 4C). Aged hens tended to have lower relative expression of *Cyp27C1* than mature hens, but no significant difference was observed. The relative expression levels (AU) of *Cyp24A1* mRNA in kidneys were 1.00 ± 0.21 in immature chicks, 0.72 ± 0.15 in mature hens, and 0.64 ± 0.21 in aged hens, showing no significant differences with maturity or aging (Figure 4D).

The relative expression levels (AU) of *Cyp27A1* mRNA in the liver were 1.00 ± 0.13 in immature chicks, 1.50 ± 0.15 in mature hens, and 1.02 ± 0.12 in aged hens, with mature hens showing significantly higher expression levels than immature chicks (Figure 4E). Aged hens showed a significant decrease in *Cyp27A1* mRNA relative expression compared to mature hens. The relative expression levels (AU) of *DBP* mRNA in the liver were 1.00 ± 0.13 in immature chicks, 3.66 ± 0.46 in mature hens, and 3.06 ± 0.48 in aged hens, showing significantly higher expression in mature and aged hens compared to immature chicks (Figure 4F). However, there was no significant difference in *DBP* mRNA expression between the mature and aged hens.

### 3.5. Localization of Meg, CUB, and DBP Proteins in Laying Hens during Maturation and Aging

The localization of the Meg and CUB proteins in the kidneys is shown in Figure 5. Meg and CUB were strongly localized in the apical membrane of the proximal tubular cells of the kidneys of immature chicks, mature hens, and aged hens. Their localization was weakly observed in the cytoplasm of proximal tubular cells. In contrast, Meg and CUB was not localized to the renal corpuscles or distal tubules. DBP protein was also localized at the apical membrane of the renal proximal tubule cells and was granular in the cytoplasm of the epithelial cells from the brush border to the cell nucleus. In particular, the localization was more pronounced in mature hens than in immature chicks, and the granular structures observed in the proximal tubular cells were large and numerous. In contrast, in aged hens, the granular structure of DBP was small and limited to the apical area.

## 4. Discussion

Previous studies have shown that in mammals, Meg and CUB are expressed mainly in the kidneys [15,22,23]. In the present study, we investigated the expression of *Meg* mRNA in mature hens, which was highest in the kidney and 700-fold higher than that in the liver. This result is similar to that reported by Plieschnig et al. (2012) in laying hens [19]. On the other hand, in this study, *CUB* mRNA expression was also the highest in the kidney, and western blot analyses showed that the expression of both proteins was observed only in the kidney. These results indicate that Meg and CUB are highly expressed in the kidneys of mature laying hens as well as in mammals. Meg and CUB are co-expressed in the apical membrane of renal proximal tubular cells, which in mammals is associated with the endocytosis of albumin, and various lipoproteins and steroid hormones filtered in the glomerulus (reabsorption) have been shown to occur [22,24]. In chickens, Meg has also been shown to localize to the apical membrane of the kidney proximal tubular cells [19]. Consistent with these reports, in the present study, we found that Meg localizes to the proximal tubular apical membrane of the kidneys of mature hens but not to the glomerulus or distal tubules. Furthermore, we are the first to show that CUB localizes to the proximal tubular apical membrane of the kidney in birds. These findings suggest that even in mature laying hens, Meg and CUB form a complex in the proximal tubules of the kidney and interact to reabsorb essential low-molecular-weight proteins filtered by the glomerulus [25,26,27].

In addition to the kidney, this study showed that *Meg* mRNA is expressed in the oviducts (magnum, isthmus, and eggshell gland) and medullary bone of mature laying hens and that *CUB* mRNA is expressed in the intestinal tract (duodenum and jejunum). In humans, in addition to Meg and CUB, 1α-hydroxylase is expressed in vitamin D target tissues, such as osteoblasts and the placenta, and the 25(OH)D_3_-DBP complex is taken up to produce 1,25(OH)_2_D_3_, indicating that 1,25(OH)_2_D_3_ regulates target tissues via an autocrine or paracrine pathway independent of circulating 1,25(OH)_2_D_3_ of kidney origin [28,29]. In laying hens, it has been demonstrated that vitamin D metabolizing enzymes, such as 1α-hydroxylase and 24-hydroxylase, are present in the eggshell gland and intestinal tract, in addition to the kidney [10,30]. Consequently, uptake of the 25(OH)D_3_-DBP complex by Meg and CUB in the intestinal tract, oviduct, and medullary bone of laying hens may regulate calcium metabolism via local vitamin D metabolism. Detailed in vitro studies on the regulation of local vitamin D metabolism in the target tissues are necessary.

Egg-laying hens require large amounts of calcium for eggshell formation, and elevated circulating 1,25(OH)_2_D_3_ concentrations have been observed [5,31,32]. Increased circulating 1,25(OH)_2_D_3_ concentrations in laying hens have been attributed to renal 1α-hydroxylase activity, which increases with maturity [5,33]. However, the increase in circulating 1,25(OH)_2_D_3_ concentrations is probably due to enhanced uptake of the 25(OH)D_3_-DBP complex by endocytic receptors, as we observed an increase in renal *Meg* and *CUB* mRNAs with maturation. In fact, microscopy indicated that DBP uptake was higher in mature hens than in immature chicks. The finding of high *DBP* mRNA expression in mature hens also suggests that 25(OH)D_3_ forms a complex with highly expressed DBP and is transported to the kidney, resulting in enhanced uptake of the 25(OH)D_3_-DBP complex by endocytosis.

In laying hens, there is a decrease in laying rate and deterioration of eggshell quality with age. Vitamin D_3_ is the major hormone regulating eggshell quality, and circulating 1,25(OH)_2_D_3_ levels decrease with age [5]. It has been assumed that these decreases in 1,25(OH)_2_D_3_ production are due to the decreased 1α-hydroxylase activity in the kidneys [5,34]. On the other hand, both Meg and CUB expression in the kidney have been shown to be essential for producing 1,25(OH)_2_D_3_, as demonstrated in Meg-deficient mice and CUB-diseased dogs [13,25]. However, the regulation of Meg and CUB expression differs among species and physiological conditions. Specifically, Meg expression is upregulated in laying hens, whereas only CUB expression is upregulated in pregnant baboons [19,35]. It has also been reported that Meg expression in the kidney decreases with age in rats, whereas CUB expression remains unchanged [36]. In this study, we found that the expression of *Meg* mRNA did not change with age in laying hens, whereas that of *CUB* mRNA significantly decreased. In addition, the endosomes of DBP in the proximal renal tubular cells of aged hens were smaller and less numerous than those in the mature hens by the present microscopy. These results indicate that CUB deficiency decreases the uptake of the 25(OH)D_3_-DBP complex and the consequent biosynthesis of 1,25(OH)_2_D_3_ in the kidney and suppresses calcium absorption in the intestinal tract, calcium reabsorption in the kidney, and eggshell formation.

Diet is commonly supplemented with 25(OH)D_3_ to prevent the deterioration of eggshell quality due to aging and heat stress, and some improvements in eggshell quality have been reported [37,38,39]. The results of this study indicate that 25(OH)D_3_ uptake is regulated by Meg and CUB. Therefore, to improve calcium metabolism, it is essential to increase the uptake of 25(OH)D_3_-DBP by endocytic receptors and metabolize 25(OH)D_3_ more efficiently to 1,25(OH)_2_D_3_. Therefore, further studies on Meg and CUB are necessary.

## 5. Conclusions

The present results indicate that high expression of Meg and CUB in the renal proximal tubules of laying hens during maturation promotes the uptake of the 25(OH)D_3_-DBP complex and its conversion to 1,25(OH)_2_D_3_, thereby regulating specific calcium metabolism in laying hens. On the other hand, we also showed that an age-related reduction in CUB expression suppresses the uptake of the 25(OH)D_3_-DBP complex in the kidney, resulting in decreased production of 1,25(OH)_2_D_3_. The novel regulatory mechanism of vitamin D metabolism in laying hens established in this study may contribute to the prevention of eggshell quality deterioration caused by aging and heat stress.

## Figures and Tables

**Figure 1 animals-14-00502-f001:**
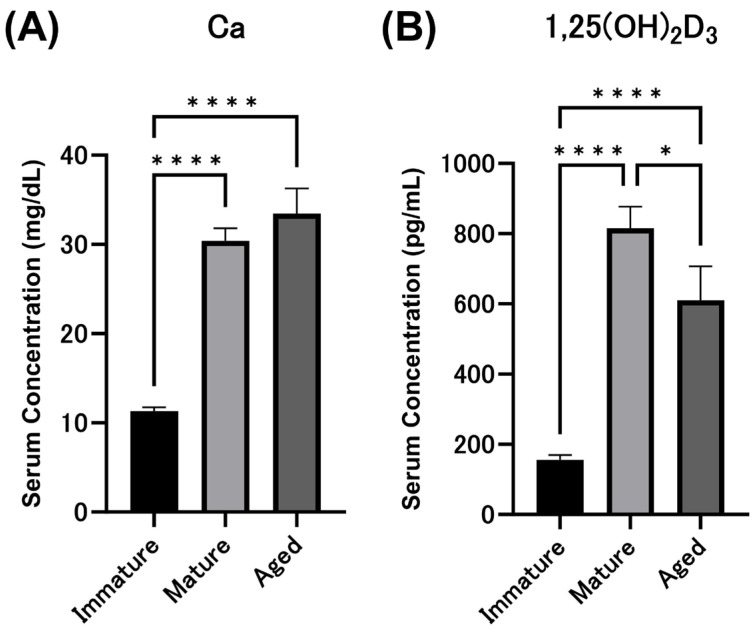
Serum calcium ((**A**) Ca) and 1,25-dihydroxy vitamin D_3_ ((**B**) 1,25(OH)_2_D_3_) concentrations. The values are the average and standard error (*n* = 7). All experiments were performed in triplicate. (* *p* < 0.05, **** *p* < 0.0001).

**Figure 2 animals-14-00502-f002:**
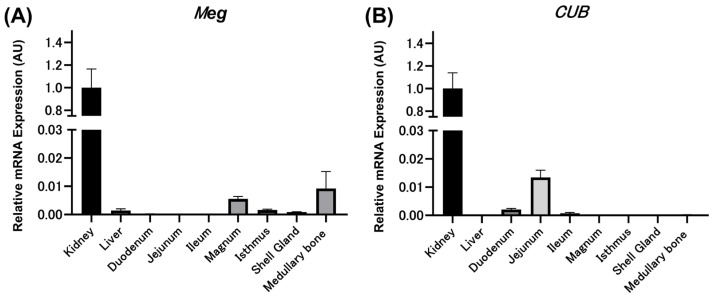
Relative expression levels (AU) of megalin ((**A**) *Meg*) and cubilin ((**B**) *CUB*) mRNAs in mature hens (*n* = 7) in each tissue. The relative expression values are represented as 1.0000 AU in the kidney. The values represent average and standard errors. All experiments were performed in triplicate.

**Figure 3 animals-14-00502-f003:**
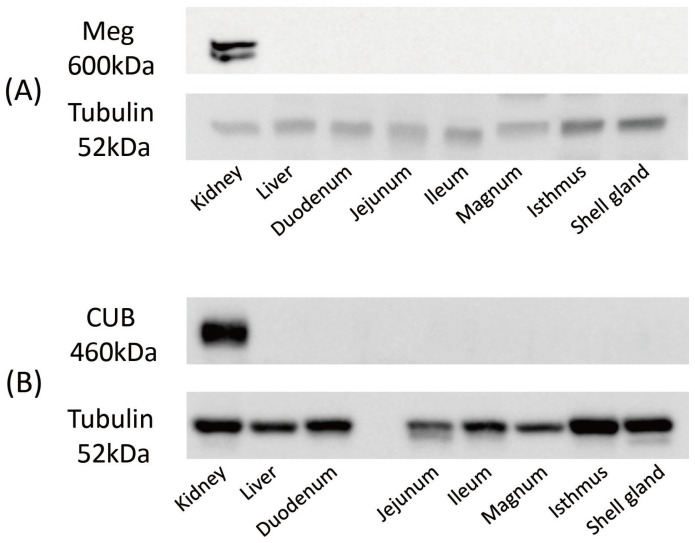
Western blotting analysis of megalin ((**A**) Meg) and cubilin ((**B**) CUB) proteins. The specific expression of Meg was observed at 600 kDa, and CUB was observed at 460 kDa. Tubulin (52 kDa) was used for loading control. All experiments were conducted with three mature hens (*n* = 3) and were repeated at least twice for each individual. The original western blot figures in Appendix A.

**Figure 4 animals-14-00502-f004:**
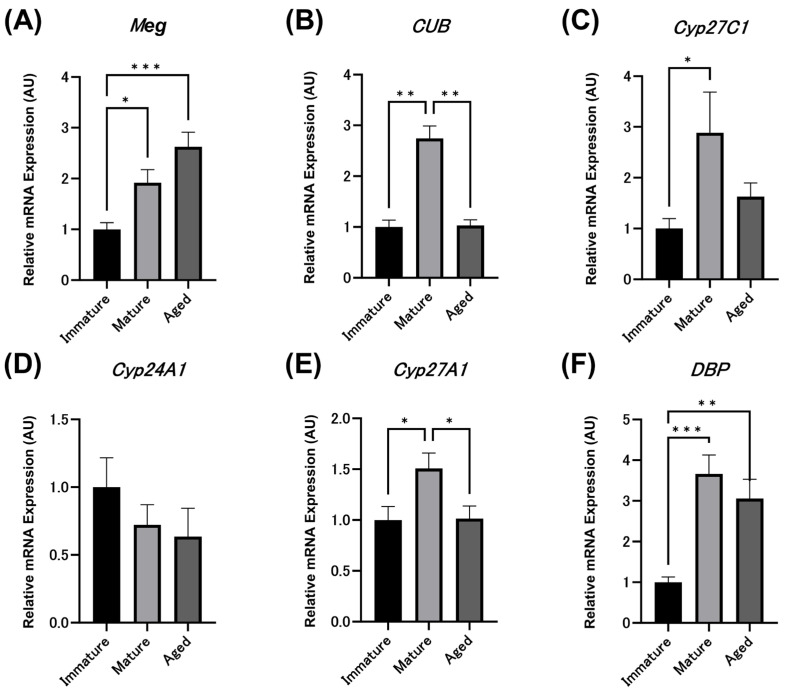
Relative expression levels (AU) of megalin ((**A**) *Meg*), cubilin ((**B**) *CUB*), 1α-hydroxylase ((**C**) *Cyp27C1*), 24-hydroxylase ((**D**) *Cyp24A1*), 25-hydroxylase ((**E**) *Cyp27A1*), and vitamin D binding protein ((**F**) *DBP*) mRNAs. The relative expression values are represented as 1.00 AU in immature hens. The values are the average and standard error (*n* = 7). All experiments were performed in triplicate. (* *p* < 0.05, ** *p* < 0.01, *** *p* < 0.001).

**Figure 5 animals-14-00502-f005:**
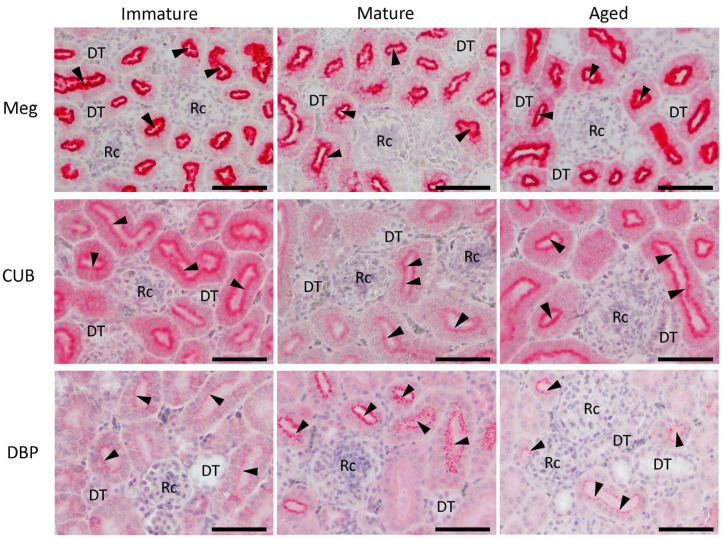
Immunohistochemical localization of megalin (Meg), cubilin (CUB), and vitamin D binding protein (DBP) in the kidneys of immature chicks and mature and aged hens. The high-density localization (observed as red) of Meg and CUB (arrowheads) was observed in the apical membranes of the proximal tubular cells of immature chicks, mature hens, and aged hens. However, no localization of either protein was observed in the renal corpuscles (Rc) or the distal tubules (DT). DBP was also localized at the apical membrane of the proximal tubular cells and was granular in the cytoplasm of epithelial cells from the brush border to the cell nucleus (arrowheads). In particular, localization was more pronounced in mature hens than in immature chicks, and the granular structures observed were large and numerous. In contrast, in aged hens, the granular structure of the DBP was small and limited to the apical area (arrowheads). Immunohistochemical observations were performed twice for all individuals (*n* = 7), and typical images are shown. PT = proximal tubule; DT = distal tubule; Rc = renal corpuscle; bars = 50 μm.

**Table 1 animals-14-00502-t001:** List of primer sequences.

Target Genes NCBI Gene Bank No.	Primer Sequence		Product Size
Megalin (*Meg*) XM_040676662.2	Forward Reverse	5′-TGCTCTGGGCATCCTTACCAC-3′ 5′-ACATTATGGGAACAACATTCTGCAC-3′	151 bp
Cubilin (*CUB*) XM_040663409.2	Forward Reverse	5′-ACAGCTGGAGGAGAGCAATG-3′ 5′-GGACAGACGAGGCTGTTCAT-3′	149 bp
*β-actin* NM_205518.2	Forward Reverse	5′-ATTGTCCACCGCAAATGCTTC-3′ 5′-AAATAAAGCCATGCCAATCTCGTC-3′	113 bp
1α-hydroxylase (*Cyp27C1*) XM_040676644.2	Forward Reverse	5′-TCGTGGCAGGAATACAGAGA-3′ 5′-ACTGCCACATCTTTGGGTTT-3′	125 bp
24-hydroxylase (*Cyp24A1*) NM_001396287.1	Forward Reverse	5′-AGCCAAGCACTCCATTGACA-3′ 5′-AACTGTTGGCCGTCGTTTCA-3′	161 bp
25-hydroxylase (*Cyp27A1*) XM_040676620.2	Forward Reverse	5′-TATCCCCAAGATGCCGATGC-3′ 5′-TGGGGAAGAGGTAGTCTCCG-3′	125 bp
VitaminD binding protein (*DBP*) NM_204882.2	Forward Reverse	5′-TGAGGACCTTTCCCCTCTCG-3′ 5′-GTGTGCTCCGACAACTTCTG-3′	102 bp

## Data Availability

The data presented in this study are available on request from the corresponding author.

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
