# Peer review of "Renal Endocytic Regulation of Vitamin D Metabolism during Maturation and Aging in Laying Hens"

_animals, 2024, doi:10.3390/ani14030502_

Round 1

Reviewer 1 Report

Comments and Suggestions for Authors

The manuscript studies the Renal Endocytic Regulation of Vitamin D Metabolism During Maturation and Ageing in Laying Hens.

However, the following observations can be made.

1. Abstract. Add a research goal. In terms of writing the research results, indicate the exact values instead of “increase”, “decrease”, etc.

2. Introduction. Write the purpose of the study.

3. Materials and methods. Move the paragraph describing statistical processing (2.4) to the end of this section. Those. it will become clause 2.6.

4. Figure 2 and 4. No units for relative expression.

5. Conclusion. Expand the description of the research results. Assess the prospects for using the results obtained in practice.

Comments on the Quality of English Language

The manuscript is written in good English.

However, the text contains minor errors and typos.

I recommend minor editing of the English of the manuscript.

Author Response

We are deeply impressed by the valuable comments of the reviewers. We would be happy to hear your decision as soon as possible.

  1. We revised the text of the abstract and clarified the goals. In addition to "increase" and "decrease," they added exact values for 1,25(OH)2D3 concentrations and mRNA relative expression (Line 29-35).

  2. We revised the text of the Introduction to clearly state the purpose of this research (Line 90-93).

  3. In the Materials and Methods, we moved the statistical processing section to the end, to section 2.6.

  4. We added AU (arbitrary units) units to Figures 2 and 4.

  5. In conclusion, we briefly describe the developments resulting from the results of this study (Line 364-366).

Reviewer 2 Report

Comments and Suggestions for Authors

In the submitted manuscript, a detailed analysis of the regulation of vitamin D metabolism at different ages of laying hens and chicks is performed. The topic of the manuscript is very relevant because vitamin D is essential for metabolism, egg quality and also for the welfare of birds.

I have the following notes for the authors:

For what period did the test groups of birds take vitamin D?

What is the concentration and chemical structure of the vitamin D in the ration, and could this affect the traits being studied?

Can the authors state with conviction that the age maturation of the birds is the factor that affects the expression of Meg and CUB and the metabolism of vitamin D, and not the differences in the feed of individual categories of birds.

Author Response

We are deeply impressed by the valuable comments of the reviewers. We would be happy to hear your decision as soon as possible.

1. The research was conducted using normal farm-raised laying hen feeding methods. We describe in our Materials and Methods (Section 2.1.) the amount of vitamin D in the diets and the duration of feeding (Line 103-108).

2. cholecalciferol. We described the conetration of vitamin D in Section 2.1. 2000IU/kg diet for immature chicks and 3300 IU/kg diet for mature and aged hens.

3. In immature chickens and laying hens, increasing vitamin D in the diet eventually leads to an upper threshold of 1,25OH2D3 concentration in the blood. This limit is significantly higher in laying hens than that in immature chicks. This indicates that the blood 1,25OH2D3 concentration depends not only on the amount of vitamin D3 in the diet but also on vitamin D metabolism in the body. In addition, a comparison of blood vitamin D3 concentrations in laying hens and aged hens reported that they were lower in aged hens, even though they were fed the same feed. Therefore, although the amount of vitamin D in the diet is important, the endocytosis potential of 25(OH)D3-DBP in the kidney may determine its ability to produce 1,25OH2D3, considering the results of this study.

Reviewer 3 Report

Comments and Suggestions for Authors

In this study, the author found that 25-hydroxyvitamin D-binding protein complex is taken up by the endocytic receptors megalin and cubilin in the renal proximal tubules, providing a novel mechanism for the regulation of vitamin D metabolism during maturation and aging. These findings may contribute to the prevention of heat stress and age-related deterioration of eggshell quality in egg laying hens. However, the readability of the manuscript can also be greatly improved. Through editing and some modifications, I think this manuscript will be more suitable for publication.

1. Lines 23-25 Meg and CUB are italicized, while other occurrences of Meg and CUB are not italicized. Please unify the full text format.

2. Should Line 42 CaCO3 numbers be subscript? Please check the entire text for subscript numbers and standardize the format.

3. All the charts in this manuscript are unclear. Please provide higher resolution charts.

4. Results section, why do many results lack units, such as lines 190-197? Please ask the author to provide a detailed explanation.

5.lines 186, 203, 248, n=7, “n” should be presented in italics.

6.There are multiple instances throughout the manuscript where there is no space between numbers and units, including lines 136, 139, 142, 143, 145, 147, and so on. The author should carefully review the entire manuscript and make the necessary modifications.

7.There are numerous instances in the manuscript where units are incorrectly written, including ml should be mL and dl should be dL, among others. The author should thoroughly check the entire manuscript for these errors and make the necessary corrections.

8.The p-values in the manuscript should be presented in italics. The author should review the entire manuscript and make the necessary changes.

9. The reference list should primarily include research publications from the past three years. However, this manuscript has a total of 32 references contains 90% of references from before 2019. Therefore, the reference significance is not significant. And the reference format is not consistent. The author should replace the references and standardize the format.

Comments on the Quality of English Language

Minor editing of English language required

Author Response

We are deeply impressed by the valuable comments of the reviewers. We would be happy to hear your decision as soon as possible.

Reviewer 3

1. Lines 23-25 Meg and CUB are italicized, while other occurrences of Meg and CUB are not italicized. Please unify the full text format.

Answer. Genes such as mRNA are italicized according to the MDPI Layout Style Guide.

2. Should Line 42 CaCO3 numbers be subscript? Please check the entire text for subscript numbers and standardize the format.

Answer. Numbers in the CaCO3, vitamin D3, 25OHD3, and 1,25OH2D3 phrases have been corrected to subscripts. We read the entire manuscript carefully and numbered the corresponding words with subscripts.

3. All the charts in this manuscript are unclear. Please provide higher resolution charts.

Answer. We redesigned the tables and figures so that they were clearly visible.

4. Results section, why do many results lack units, such as lines 190-197? Please ask the author to provide a detailed explanation.

Answer. We analyzed the changes in mRNA expression in a given sample relative to another reference sample (kidney in Figure 2 and immature chicks in Figure 4). Therefore, the relative expression levels of each gene in the kidney (Figure 2) and immature chicks (Figure 5) were expressed as value 1.000 AU (arbitrary units) normalized by the internal control gene (β-actin, which is stable and does not change in response to function). We have described these definitions in line Line133-135.

5. lines 186, 203, 248, n=7, “n” should be presented in italics.

Answer. We have corrected the "n" to be in italics.

6. There are multiple instances throughout the manuscript where there is no space between numbers and units, including lines 136, 139, 142, 143, 145, 147, and so on. The author should carefully review the entire manuscript and make the necessary modifications.

Answer. We have followed your suggestion and inserted spaces between numbers and units. We have also carefully reviewed the entire manuscript and made corrections.

7. There are numerous instances in the manuscript where units are incorrectly written, including “ml” should be “mL” and “dl” should be “dL”, among others. The author should thoroughly check the entire manuscript for these errors and make the necessary corrections.

Answer. We have corrected the error in the units you indicated and have carefully reviewed and corrected the entire manuscript.

8. The p-values in the manuscript should be presented in italics. The author should review the entire manuscript and make the necessary changes.

Answer. We have corrected italics on p-values. We carefully reviewed and corrected the entire manuscript.

9. The reference list should primarily include research publications from the past three years. However, this manuscript has a total of 32 references contains 90% of references from before 2019. Therefore, the reference significance is not significant. And the reference format is not consistent. The author should replace the references and standardize the format.

Answer. We added seven citations (2020-2023) and replaced older references with newer ones. However, we kept the citations that were particularly important in this research area. We also reviewed and corrected the formatting.

Reviewer 4 Report

Comments and Suggestions for Authors

Renal Endocytic Regulation of Vitamin D Metabolism During Maturation and Ageing in Laying Hens

Dear Authors,

The manuscript is very interesting and well prepared. Regulation of vitamin D levels in laying hens plays a very important role in the quality of the eggshell, as well as in the strength of their skeleton during the increased demand for calcium, especially in hens such as Leghorns. There are not much to correct in the text of manuscript.

Below I add some suggestions helpful in this process:

Line 20

In the text of manuscript is: ‘…the vitamin D metabolite (25OHD3) produced in liver…’, Maybe better will be to change for: ‘…the vitamin D metabolite: (25(OH)D3)…’ .

Line 23

Same like in line 20. In text is: ‘…and is metabolized to 1,25 dihydroxyvitamin D3 (1,25OH2D3)…’, can be changed for: and is metabolized to 1,25 dihydroxyvitamin D3 (1,25(OH)2D3)

Line 24

1,25(OH)2D3

Line 33

25(OH)D3-DBP

1,25(OH)2D3

And so on to line 334

Line 42

CaCO3 should be CaCO3

Line 124

Maybe list of primer sequences could be copied to line 112 to decrease space between line 124 and 125

Line 130

Small letter p must be used (sample).

Line 184

Labels will be useful to read values from Figure 1 and maybe using letter description of significance at p<0.05/0.01 (as described in 2.4. Statistical analysis subsubsection) with  superscripts (a,b… or A,B,…), will be more legible and will describe significance level between all 3 treatments.

Line 200

Labels will be useful to read values from Figure 2, maybe it is possible to decrease number of decimals describing mean and standard deviation.

Line 243

Labels will be useful to read values from Figure 3. Maybe use letter description of significance at p<0,05/0,01 with  superscripts (a,b… or A,B,…) will be more readable and will describe significance level between all 3 treatments.

Author Response

Author’s response to Reviewer 4

We are deeply impressed by the valuable comments of the reviewers. We would be happy to hear your decision as soon as possible.

Reviewer 4

Line 20

In the text of manuscript is: ‘…the vitamin D metabolite (25OHD3) produced in liver…’Maybe better will be to change for: ‘…the vitamin D metabolite: (25(OH)D3)…’ .

Answer. We have followed your suggestion and changed the description to 25(OH)D3.

Line 23

Same like in line 20. In text is: ‘…and is metabolized to 1,25 dihydroxyvitamin D3 (1,25OH2D3)…’, can be changed for: and is metabolized to 1,25 dihydroxyvitamin D3 (1,25(OH)2D3)

Answer.We have followed your suggestion and changed the description to 1,25(OH)2D3.

Line 24

1,25(OH)2D3

Answer. We have followed your suggestion and changed the description to 1,25(OH)2D3.

Line 33

25(OH)D3-DBP

1,25(OH)2D3

And so on to line 334

Answer. We have followed your suggestion and changed the description to 25(OH)D3-DBP and 1,25(OH)2D3.

Line 42

CaCO3 should be CaCO3

Answer. We have followed your suggestion and changed the description to CaCO3.

Line 124

Maybe list of primer sequences could be copied to line 112 to decrease space between line 124 and 125

Answer. The style has been shifted due to corrections to the text and figures. We would like to leave the style of the paper to the editorial staff.

Line 130

Small letter p must be used (sample).

Answer. We used small letter p as you suggested.

Line 184

Labels will be useful to read values from Figure 1 and maybe using letter description of significance at p<0.05/0.01 (as described in 2.4. Statistical analysis subsubsection) with superscripts (a,b… or A,B,…), will be more legible and will describe significance level between all 3 treatments.

Line 200

Labels will be useful to read values from Figure 2, maybe it is possible to decrease number of decimals describing mean and standard deviation.

Line 243

Labels will be useful to read values from Figure 3. Maybe use letter description of significance at p<0,05/0,01 with superscripts (a,b… or A,B,…) will be more readable and will describe significance level between all 3 treatments.

Answer. P value has increased due to a change in the statistical processing method. We appreciate your valuable suggestion, but we have listed it as it is now.

Reviewer 5 Report

Comments and Suggestions for Authors

Authors have provided an interestign manuscript that provides new information on the literature.

New references especially on heat stress on layer hens and vitamin D usage in layers and poultry are needed.

LInes 65-76, can be reduced as they refer to mammals.

In material and methods, experimental design should be enlarged. How many animals per group? what was their feeding?

In methodology, authors should provide the inhouse and control gene to compare the gene expression. Control measurements and gene banks should be provided in detail. 

Determination of calcium by kit is not the most reliable way to measure it. Authors must provide recovery and accuracy data.

Atuhors must explain if one way anova is the best way to analyse gene expression. In my opinion, other methodology should extra included.

For histochemistry, did the authors analyse their data in quantitative or semiquantitative way?

In histology and histochemistry, number of samples evaluated should be written. What was the n- replicate in each analysis? This should be written in every table- figure.

One reference number is in red, why?

Some recent references 2021-2023 should be added.

Minor points (CaCO3); & D3;1,25OH2D3) need to have numbers as indices and not normal numbers, throughout.

Comments on the Quality of English Language

The use of English is acceptable. Only minor points should be checked.

Author Response

We are deeply impressed by the valuable comments of the reviewers. We would be happy to hear your decision as soon as possible.

Reviewer 5

1. New references especially on heat stress on layer hens and vitamin D usage in layers and poultry are needed.

Answer. We have added citations on heat stress and dietary vitamin D.

2. Lines 65-76, can be reduced as they refer to mammals.

Answer.  To date, there have been few publications on megalin and cubilin except for those on mammals. Since the functions of megalin and cubilin revealed by mammalian studies form the basis of this study, we have left the text as it is.

3. In material and methods, experimental design should be enlarged. How many animals per group? what was their feeding?

Answer.  We have added the matters you have mentioned about experimental design (Line 103-108).

4. In methodology, authors should provide the inhouse and control gene to compare the gene expression. Control measurements and gene banks should be provided in detail. 

Answer. The information in the Genebank was reconfirmed and corrected (Table 1). The internal control of qPCR was described in Materials and Methods. The qPCR was performed according to the MIQE guideline for qPCR (Bustin et al., 2009).

Bustin et al (2009) Clinical Chemistry, 55: 611-622.

5. Determination of calcium by kit is not the most reliable way to measure it. Authors must provide recovery and accuracy data.

Answer.  This kit is reliable using the methyl xylenol blue method and has been used to measure calcium levels in many papers listed below. Unfortunately, we cannot show the recovery rate, but in this study, we used standad to construct a calibration curve and properly measure calcium concentration.

  1. Tanikake et al. (2017) Cell Transplantation, 26: 1066-1076.
  2. Inamoto et al. (2023) Journal of Dental Sciences, 18: 1079-1085.
  3. Tsuda et al. (2020) Journal of Pharmacological Sciences, 142: 109-115.

6. Authors must explain if one way ANOVA is the best way to analyze gene expression. In my opinion, other methodology should extra included.

Answer.  Gene expression analysis was re-examined using the Neuman-Keuls method (Line 185-187), which is used in many papers described below, and the results were described. The results section has been partially revised (Line198 and Line 234-258), and Figures 1 and 3 have been corrected to reflect the change in significance.

  1. Zheng et al. (2024) Brain Research Bulletin, 207: 110871.
  2. Yanez-Mendizabal et al. (2023) Biological Control, 185: 10536.
  3. Heredia-Garcia, et al. (2023) Chemosphere, 330: 138729.
  4. Reference [10] in this manuscript

7. For histochemistry, did the authors analyse their data in quantitative or semiquantitative way? In histology and histochemistry, number of samples evaluated should be written. What was the n- replicate in each analysis? This should be written in every table- figure.

Answer. We have not performed quantitative experiments on immunohistochemical observations. However, we have performed immunohistochemical observations on all individuals twice, and we recognize the histological features described in the results. Figure 5 shows a typical image. For the other figures and tables, the number of samples and the number of experiments are listed.

8. One reference number is in red, why?

Answer.  This is our error.This is our error. We have corrected it.

9. Some recent references 2021-2023 should be added.

Answer. We added seven citations (2020-2023) and replaced older references with newer ones. However, we kept the citations that were particularly important in this research area.

10. Minor points (CaCO3); & D3;1,25OH2D3) need to have numbers as indices and not normal numbers, throughout.

Answer. We have corrected the numbers in the terms you mentioned to be subscripted.

Round 2

Reviewer 2 Report

Comments and Suggestions for Authors

I agree all of answers from the authors. This is logical and high scientific level response. 

Reviewer 5 Report

Comments and Suggestions for Authors

Authors have revised their work adequately.